# Augmenting Ontology Alignment by Semantic Embedding and Distant Supervision

Jiaoyan Chen[1], Ernesto Jiménez-Ruiz[2,3], Ian Horrocks[1], Denvar Antonyrajah[4], Ali Hadian[4], and Jaehun Lee[5]

[1] Department of Computer Science, University of Oxford, UK
[2] City, University of London, UK
[3] SIRIUS, University of Oslo, Norway
[4] Samsung Research, UK
[5] Samsung Research, Korea

**Abstract.** Ontology alignment plays a critical role in knowledge integration and has been widely investigated in the past decades. State of the art systems, however, still have considerable room for performance improvement especially in dealing with new (industrial) alignment tasks. In this paper we present a machine learning based extension to traditional ontology alignment systems, using distant supervision for training, ontology embedding and Siamese Neural Networks for incorporating richer semantics. We have used the extension together with traditional systems such as LogMap and AML to align two food ontologies, HeLiS and FoodOn, and we found that the extension recalls many additional valid mappings and also avoids some false positive mappings. This is also verified by an evaluation on alignment tasks from the OAEI conference track.

**Keywords:** Ontology Alignment · Semantic Embedding · Distant Supervision · Siamese Neural Network

## 1  Introduction

Ontologies are widely used to represent, manage and exchange (domain) knowledge. However, the content of any single ontology is often incomplete even in a single domain and, moreover, many real world applications rely on cross-domain knowledge. Integration of multiple ontologies is therefore a critical task, and is often implemented by identifying cross-ontology mappings between classes that have an equivalent- or sub-class relationship. This process is known as ontology alignment or ontology matching [29,8,26].[6]

Ontology alignment has been been investigated for many years. State of the art (SOTA) systems such as LogMap [17,18] and AgreementMakerLight (AML) [9] often combine multiple strategies such as lexical matching, structural matching and logical reasoning. Such systems typically use lexical matching as their

---

[6] Ontology alignment also includes mappings between individuals and properties, as well as mappings with more complicated relationships beyond atomic subsumption and equivalence. In this study we focus on mappings between equivalent classes.

starting point, and while this captures string or token similarity, it fails to capture the contextual meaning of words. Logical reasoning can be used to improve mapping quality, but this often wrongly rejects some valid mappings [27]. In practice, such systems often need (combinations of) hand-craft matching methods to achieve good performance for a new task.

The last decade has seen an extensive investigation of semantic embedding, a branch of machine learning (ML) techniques which can encode symbols such as natural language words, ontology concepts, knowledge graph entities and relations into vectors with their semantics (*e.g.*, correlation with the neighbours) [23,31,20]. This enables us to augment the aforementioned ontology alignment systems with ML algorithms that can exploit richer semantics so as to recall some missed mappings and avoid some false positives.

In this paper we present a ML extension that utilizes distant supervision and semantic embedding, and that can be used to augment classic ontology alignment systems. Briefly, it first uses the original ontology alignment system plus class disjointness constraints (as heuristic rules) to generate high precision seed mappings, and then uses these mappings to train a Siamese Neural Network (SiamNN) for predicting cross-ontology class mappings via semantic embeddings in OWL2Vec*— an ontology tailored language model [3]. We have tested our ML-augmentation with the SOTA systems LogMap and AML in a real world ontology alignment task identified by our industrial partner Samsung Research UK, *i.e.*, the alignment of two food ontologies: HeLiS [7] and FoodOn [6]. The augmentation improved recall by more that 130% while at the same time achieving small improvements in precision. Smaller but still significant improvements in precision and recall were also achieved on an alignment task from the Ontology Alignment Evaluation Initiative (OAEI) [1].

In the remainder of this paper we use LogMap as a concrete example of our ML extension, but the extension can be directly applied to AML and to any other system that is capable of generating high precision mappings to be used in the training phase.

## 2    Preliminaries and Related Work

### 2.1    LogMap

LogMap is a scalable logic-based ontology matching system [17,18]. It is often one of the best performing systems for real-world tasks such as those in the biomedical tracks of the OAEI [1].

Fig. 1 shows the procedure followed by LogMap to compute an alignment $\mathcal{M}$ given two input ontologies $\mathcal{O}_1$ and $\mathcal{O}_2$. LogMap first builds a lexical index for each ontology based on its entity labels and label variations (*e.g.*, synonyms). These indexes are used to efficiently computing an over-estimation $\mathcal{M}_o$ of the mappings between $\mathcal{O}_1$ and $\mathcal{O}_2$. Mappings in $\mathcal{M}_o$ are not necessarily correct, but they link lexically-related entities and usually have a high recall, while still representing a manageable subset of all possible mappings (i.e., the Cartesian product of the sets of classes in the input ontologies) [16].

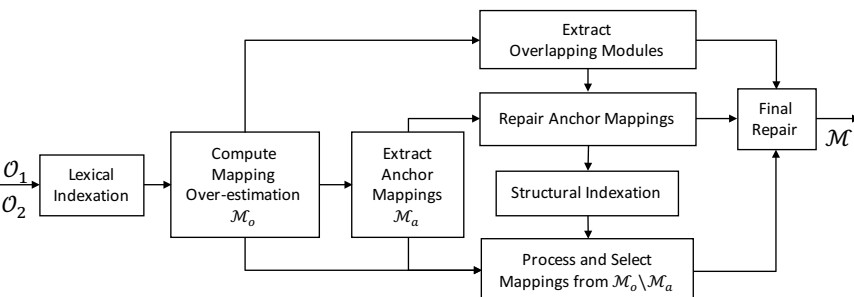

Fig. 1: LogMap system in a nutshell [18]

From the mapping over-estimation $\mathcal{M}_o$, LogMap identifies a number of high-confidence mappings called *anchor mappings* ($\mathcal{M}_a$). These mappings are used to assess the structural and logical compatibility of the remaining candidate mappings in $\mathcal{M}_o$ via a structural index, which significantly reduces the cost of answering taxonomic and disjointness queries. Mappings in $\mathcal{M}_o$ are also assessed according to the lexical similarity of the involved entities. Finally, LogMap outputs a set of selected mappings $\mathcal{M}$ between $\mathcal{O}_1$ and $\mathcal{O}_2$, and additionally gives as output the anchor mappings $\mathcal{M}_a$ and the mapping over-estimation $\mathcal{M}_o$.

### 2.2   Machine Learning for Ontology Alignment

Machine learning (ML) has recently been explored for ontology alignment. ML should facilitate the exploitation of both class information (*e.g.*, names and annotations) and their context in an ontology (*i.e.*, their semantics). However, to develop a robust model, several critical issues have to be addressed. Next we will discuss these issues and current solutions, and compare our method with them.

**Features.** The symbolic information of a class, such as its textual label and neighbourhood graph structure, should be transformed into real values before they can be utilized by ML algorithms, and informative variables (*i.e.*, features) should be extracted to achieve high performance. One solution, as implemented in, *e.g.*, GLUE [5] and POMap++ [21], is extracting pre-defined features such as edit distance between labels and syntactic similarity. Another solution, as adopted by, *e.g.*, Zhang et al. [33], DOME [13], OntoEmma [30], Bento et al. [2], DeepFCA [22] and VeeAlign [15], is learning relevant features via representation learning models such as neural networks, or/and using pre-trained semantic embeddings (*i.e.*, vectors of characters, words or classes with their semantics kept in the vector space). Some methods such as [25] and [30] adopt both pre-defined features and semantic embeddings. Meanwhile, instead of word embeddings pre-trained by an external corpus, tailored character embeddings or document embeddings have been explored. ERSOM [32] learns ontology tailored word embeddings via an Auto-Encoder and a similarity propagation method with the classes'

meta information and context, while DeepAlignment [19] extends ERSOM by incorporating the synonymy and antonymy word relationships.

**Samples.** With semantic embeddings, we can identify mappings by calculating the vector distance (or similarity) between classes as in [32,19]. However, such an unsupervised approach depends on how the semantic embeddings are learned. For ontology-tailored embeddings, which usually achieve better performance than pre-trained embeddings, the vector spaces of the two to-be-aligned ontologies are independent and thus two equivalent cross-ontology classes may still have a large vector distance. To address the above issue and further improve the performance, we can utilize (semi-)supervised ML solutions which rely on labeled mappings (*i.e.*, samples) to learn features and train models to predict mappings. Besides costly human annotation for the training samples, one ML solution is transfer learning between tasks, *i.e.*, re-using known mappings from other aligned ontologies. For example, OntoEmma [30] trains the model using $50,523$ positive mappings between the ontologies of the Unified Medical Language System, while [25] transfers samples from tasks of one OAEI track to another. However, the effectiveness of such sample transfer significantly depends on the sources to be transferred, and it may be hard to find a suitable source for a new alignment task. In practice, neither [30] nor [25] outperformed the classic systems such as AML and LogMap on the evaluated OAEI tracks.

**Scalable Mapping Prediction.** Unlike ontology alignment systems based on lexical indexes, an ML-based method usually needs to predict or calculate the scores of all cross-ontology class pairs, which can lead to scalability problems with large ontologies. One solution for this issue is to use blocking techniques such as locality-sensitive hashing [11] and embedding-based lexical index clustering [16]. Another solution, adopted by ERSOM [32] and DeepAlignment [19], is to use optimized search algorithms such as Stable Marriage. The prediction model can also be deployed together with a set of traditional alignment systems, the union of whose outputs can act as a reduced set of candidates with a good recall [25], or with some logic-based constraints or rules which can filter out some candidates and reduce the search space.

In our ML extension, we addressed the feature issue via the combination of an ontology embedding method named OWL2Vec* [3], which is a neural language model tailored to the ontology's text, graph structure and logical axioms; and a SiamNN, which learns features of the input classes and bridges the gap between the two embedding spaces. Unlike current (semi-)supervised learning methods, our ML extension addresses the sample shortage issue via a distant supervision strategy, where some confident mappings derived by the to-be-extended system (such as the anchor mappings of LogMap) are used to generate positive and negative samples, with some high-level class disjointness constraints used to improve sample quality. To reduce the search space when aligning large ontologies, we can optionally use the to-be-extended system (or some other traditional system) to compute a set of candidate mappings with very high recall (similar to [25]); in the case of LogMap we can use its so-called mapping over-estimation.

## 3   Use Case

**Ontologies.** In this section we present the use case of aligning HeLiS[7] [7] and FoodOn[8] [6] — two large OWL[9] ontologies. FoodOn captures detailed food knowledge and other knowledge from relevant domains such as agriculture, chemistry and environment, with 359 instances, 28,182 classes and 241,581 axioms within the description logic (DL) $\mathcal{SRIQ}$. HeLiS captures general knowledge on both food and healthy lifestyles with 20,318 instances, 277 classes and 172,213 axioms within the DL $\mathcal{ALCHIQ(D)}$. In order to facilitate alignment, HeLiS instances were transformed into classes, with associated *rdf:type* triples being transformed into *rdfs:subClassOf* triples. This transformation changes the ontology's semantics, but the ontology integration still supports the knowledge graph construction application in industry and does not impact the evaluation of different systems.

A fragment of HeLiS and FoodOn is shown in Fig. 2, where each class is represented by a short name/label for readability.

**Motivations.** By providing more complete and fine-grained knowledge covering both food and lifestyles, an alignment of HeLiS and FoodOn can be used to improve personalisation and can benefit popular applications in areas such as sport, health and wellbeing. The alignment can also be used for ontology quality assurance (QA) by identifying missing and logically inconsistent relationships through cross checking. One QA example is discovering the missing subsumption relationship between "Soybean Milk" and "Soybean Food Product" in FoodOn (where "Soybean Milk" is only categorized as "Beverage") by mapping them to their HeLiS counterparts "SoyMilk" and "SoyProducts" whose subsumption relationship is defined. We have identified more than 500 such new subsumption relationships between FoodOn classes through aligning FoodOn and HeLiS [14].

**Challenges.** The technical challenges of aligning HeLiS and FoodOn lie in several aspects. First, as in many ontology matching tasks, we need to address the ambiguity between classes with similar names or with similar neighbourhood structures, the logical inconsistency that can be caused by mappings, the very large search space (with over 580 million candidate mappings), and so on. Second, FoodOn is itself composed of multiple source ontologies, including NCBITaxon and The Environment Ontology, and thus its class hierarchy includes branches covering not only food categorization but also food source categorization (closely related to biological taxonomy), chemical element categorization, etc. Similarly HeLiS has branches of nutrients, food and so on. Similar names and local contexts (*e.g.*, of food products and food sources) can lead to "branch conflicting" mappings whose classes lie in branches with different meanings. One example is the incorrect mapping between "Caesar's Mushrooms" of HeLiS (a food) and "Caesar's Mushroom" of FoodOn (a food source) as illustrated in Fig. 2. Classic

---

[7] HeLiS project: `https://horus-ai.fbk.eu/helis/`

[8] FoodOn project: `https://foodon.org/`

[9] Web Ontology Language: `https://www.w3.org/TR/owl-features/`

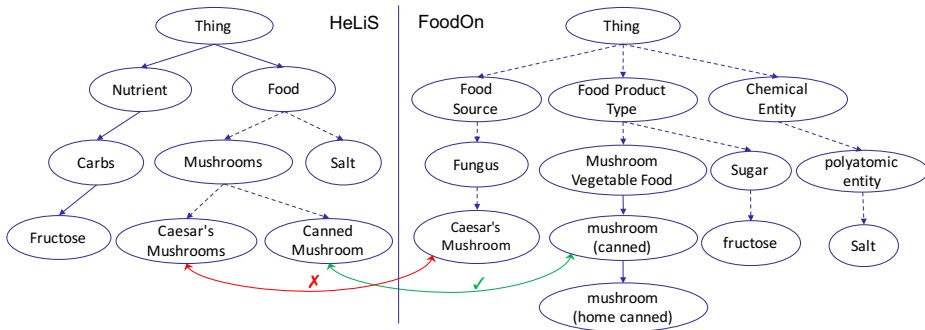

Fig. 2: Fragments of the HeLiS and FoodOn ontologies. The dash arrow means some intermediate classes are hidden. The red (green resp.) arrow denotes false (true resp.) mappings.

systems often fail to identify such errors, even when using logical assessment (as in LogMap) due to missing class disjointness axioms in the source ontologies.

## 4   Methodology

We will present our ML extension w.r.t. LogMap, as shown in Fig. 3. It comprises three steps: *(i)* compute the seed mappings starting from a set of high precision mappings (such as LogMap's "anchor" mappings) and applying class disjointness constraints (branch conflicts) to further improve precision; *(ii)* construct samples and train a mapping prediction model (a SiamNN whose input is a pair of classes or their associated paths); and *(iii)* compute the output mappings, (optionally) starting from a set of high recall candidate mappings (such as LogMap's over-estimation mappings) to reduce the search space. Note this extension can be used with any "traditional" system that is capable of generating high precision mappings for use in the training phase (in our evaluation we use AML as well as LogMap).

### 4.1   Seed Mappings

To achieve high-confidence seed mappings ($\mathcal{M}_s$) for training, we define a set of disjointness constraints between cross ontology classes to filter out some false-positive mappings from the LogMap anchor mappings ($\mathcal{M}_a$). A disjointness constraint is denoted $\delta = (c_1, c_2)$, where $c_1$ and $c_2$ are typically very general classes in $\mathcal{O}_1$ and $\mathcal{O}_2$ respectively, acting as the "root" classes of different knowledge branches. For example, in Fig. 2, "Food" of HeLiS and "Food Source" of FoodOn comprise one disjointness constraint, while "Food" of HeLiS and "Chemical Entity" of FoodOn comprise another constraint. The set of constraints, which we denote $\Delta$, together with the original alignment system act as heuristic rules in

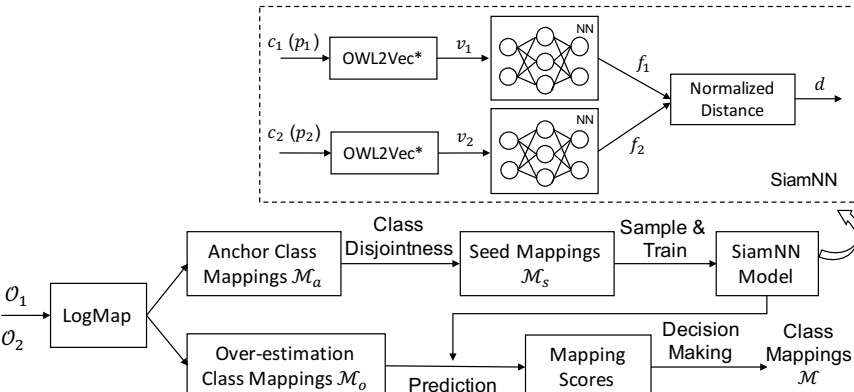

Fig. 3: The ML extension for LogMap

normal distant supervision. In our HeLiS-FoodOn case study, we manually defined four disjointness constraints based on knowledge of the domain and the ontology architectures. Given that disjointness constraints typically involve only very general classes, defining them does not require very detailed knowledge of the domain; moreover, we can use statistical analysis of the mappings computed by LogMap to identify candidate disjointness constraints, or even to fully automate the definition of $\Delta$. For example, given two sibling classes $c$ and $c'$ in $\mathcal{O}_1$ and a class $s$ in $\mathcal{O}_1$, if there are $n$ mappings from subclasses of $c$ to subclasses of $s$ and $n'$ mappings from subclasses of $c'$ to subclasses of $s$, with $n' \ll n$, then $c'$ is likely to be disjoint with $s$ and $(c', s)$ can be used as a (candidate) disjointness constraint.

When using $\Delta$ to filter a mapping $m = (c_1, c_2) \in \mathcal{M}_a$, we consider not just $c_1$ and $c_2$, but all subsumers of $c_1$ and $c_2$ in the corresponding ontologies $\mathcal{O}_1$ and $\mathcal{O}_2$. For this purpose we use the OWL reasoner HermiT [12] to compute the set of subsumers (both explicitly asserted and entailed) of a given class $c$, which we denote $P_c$. Then, given a mapping $m = (c_1, c_2) \in \mathcal{M}_a$, we discard $m$ as a false-positive if there is some $(c_1', c_2') \in \Delta$ such that $c_1' \in P_{c_1}$ and $c_2' \in P_{c_2}$; if this is *not* the case, then we add $m$ to the set of seed mappings $\mathcal{M}_s$.

### 4.2   Siamese Neural Network

We first generate positive and negative class mappings (samples), then embed these samples as vectors using OWL2Vec$^*$, and finally we train a SiamNN as the mapping prediction model. The seed mappings $\mathcal{M}_s$ are adopted as the positive samples and are randomly divided into a training set $\mathcal{M}_s^t$ and a validation set $\mathcal{M}_s^v$ by a given ratio $\gamma$. We then generate the corresponding negative sample sets of $\mathcal{M}_s^t$ and $\mathcal{M}_s^v$, denoted as $\mathcal{M}_s^{t'}$ and $\mathcal{M}_s^{v'}$ respectively, as follows. For each mapping $m = (c_1, c_2)$ in $\mathcal{M}_s^t$ or $\mathcal{M}_s^v$, we generate one negative sample $m'$ by replacing $c_1$ with a class $c_1'$ randomly selected from $\mathcal{O}_1$, and we generate a second

negative sample $m''$ by replacing $c_2$ with a class $c_2'$ randomly selected from $\mathcal{O}_2$. Note that the random replacements could produce positive samples from $\mathcal{M}_s$; we discard any such negative samples. We also adopt those anchor mappings that violate the class disjointness constraints, $i.e.$, $\mathcal{M}_a \setminus \mathcal{M}_s$, as negative samples, and randomly partition them into a training set $\mathcal{M}_a^{t'}$ and a validation set $\mathcal{M}_a^{v'}$ with the same ratio of $\gamma$. We finally get the training samples as a tuple of $\mathcal{M}_s^t$ and $\mathcal{M}_s^{t'} \cup \mathcal{M}_a^{t'}$ to train the SiamNN, and the validation samples as a tuple of $\mathcal{M}_s^v$ and $\mathcal{M}_s^{v'} \cup \mathcal{M}_a^{v'}$ to adjust the hyper parameters such as the network architecture and the embedding option.

The OWL2Vec* embedding of an ontology is a language model tailored to the given ontology. It can be first pre-trained with a large normal text corpus such as Word2Vec and then fine-tuned with a corpus whose sequences include walks over the ontology's graph structure, the ontology's axioms, the ontology's textual information ($e.g.$, class labels, definitions and comments), etc. It can also be directly trained with the ontology's corpus. In this study, we evaluated both training settings. The OWL2Vec* embedding encodes a class in two ways: directly adopting the vector of the class's URI or calculating the average word vector of the words of the class labels. We prefer the latter as it can utilize both pre-training and fine-tuning, and often performs better for ontologies with rich textual information such as FoodOn. Please refer to [3] for more details on OWL2Vec*. Given a class $c$, we denote its OWL2Vec* embedding as $v(c)$.

For each mapping sample $m = (c_1, c_2) \in \mathcal{M}_s^t$, we consider two kinds of embeddings to transform it into a tuple composed of two vectors. The first option is directly adopting its OWL2Vec* embeddings $v(c_1)$ and $v(c_2)$, $i.e.$, $v(m) = \{v(c_1), v(c_2)\}$. The second option is to augment the context of $c_1$ and $c_2$ by embedding the associated paths of $c_1$ and $c_2$, $i.e.$, the sequences of classes obtained by traversing the class hierarchy back to $owl{:}Thing$. As one class may have multiple such paths, we randomly select at most two paths for each class, and thus one class mapping sample leads to at most four path mapping samples. For two paths associated with $c_1$ and $c_2$, denoted as $p_1 = (c_1, ..., c_{n_1}) \in P_1$ and $p_2 = (c_2, ..., c_{n_2}) \in P_2$ respectively, the mapping embedding $v(m)$ is calculated as either $\left\{ \frac{1}{n_1} \sum_{c_i=c_1}^{c_{n_1}} v(c_i), \frac{1}{n_2} \sum_{c_j=c_2}^{c_{n_2}} v(c_j) \right\}$ or $\{[v(c_1), ..., v(c_{n_1})], [v(c_2), ..., v(c_{n_2})]\}$, where $[\cdot, \cdot]$ denotes the vector concatenation. The former embeds a path by averaging the embeddings of its classes, while the latter embeds a path by concatenating the embeddings of its classes. In a given ontology, different paths can have different lengths. To align the vectors of different samples, we fix the path length for the ontology by setting it to the length of the longest path, and pad the shorter paths with placeholders whose embeddings are zero vectors. Note the path lengths of $\mathcal{O}_1$ and $\mathcal{O}_2$ can be different.

The SiamNN is composed of two networks that have the same architecture. The two vectors of an input mapping are fed into the two networks respectively, two features (denoted as $f_1$ and $f_2$) are then calculated accordingly, and their normalized distance $d = \frac{\|f_1 - f_2\|}{\|f_1\| + \|f_2\|}$ is further calculated as the output, where $\|\cdot\|$ denotes the Euclidean norm of a vector. A smaller distance indicates that the two classes corresponding to the two input vectors are more likely to constitute

a valid mapping, and vice versa. The two networks are learned together by minimizing the following contrastive loss using the Adam optimizer:

$$Loss = \sum_{i=0}^{N} \frac{y_i \times d_i + (1 - y_i) \times max\{\epsilon - d_i, 0\}}{2} \quad (1)$$

where $i$ denotes the $i^{\text{th}}$ mapping sample, $d_i$ denotes its output distance, $y_i$ denotes its label ($y_i$=1 if the mapping is positive, $y_i$=0 otherwise), $N$ denotes the sample number and $\epsilon$ denotes a margin value. Note the insight behind the SiamNN is to map the two input vectors into the same space via two networks (non-linear transformations) which at the same time learn features.

Different network architectures can be adopted for feature learning. We evaluated one simple network (Multi-layer Perception (MLP) with two hidden layers), one classic sequence learning model (Bidirectional Recurrent Neural Networks (BiRNN) with Gate Recurrent Units [4]), and the BiRNN with an additional attention layer (AttBiRNN).

### 4.3  Prediction, Filtering and Ensemble

We can simply consider all cross-ontology class pairs to be candidate mappings, but this leads to a very large number if the ontologies are large. In our extension to LogMap, we adopt its over-estimation class mappings $\mathcal{M}_o$ as the candidates. This reduces the potential number of candidates from around 580 million to 8,891 when aligning HeLiS and FoodOn, and at the same time helps to avoid false positive mappings. Each mapping $m = (c_1, c_2) \in \mathcal{M}_o$ is embeded into a tuple of vectors in the same way as in training the SiamNN except that only one path is randomly selected for each class if path embedding is adopted. A distance $d \in [0, 1]$ is then predicted by the SiamNN, and a score $y$ is further calculated as $1 - d$. A higher score indicates a more likely mapping, and vice versa. A score threshold $\theta$ is used to filter out unlikely mappings: $m$ is accepted if $y \geq \theta$, and rejected otherwise. To determine $\theta$, we utilized the validation samples: the threshold is increased from 0 to 1 with a small step (*e.g.*, 0.02), and the value leading to the best performance (*e.g.*, the highest F1 score in aligning HeLiS and FoodOn) on $\mathcal{M}_s^v$ and $\mathcal{M}_s^{v'} \cup \mathcal{M}_a^{v'}$ is adopted. The resulting class mappings are denoted as $\mathcal{M}_p$ (predicted mappings).

We filter out any predicted class mappings in $\mathcal{M}_p$ that violate class disjointness constraints $\Delta$ as when generating seed mappings (see Section 4.1). We further filter the mappings in $\mathcal{M}_p$ using a subsumption-based logical assessment. Specifically, a class in one ontology cannot be equivalent to multiple classes that are in a sub-class relationship in the other ontology. For example, "Canned Mushroom" of HeLiS cannot be equivalent to both "mushroom (canned)" and "mushroom (home canned)" in FoodOn in Fig. 2. If this happens, then the mapping with a lower prediction score should be discarded. More formally, if $\mathcal{M}_p$ includes two mappings $(c_1, c_2)$ and $(s_1, s_2)$ such that either $c_1 = s_1$ and $c_2$ subsumes $s_2$ in $\mathcal{O}_2$, or $c_2 = s_2$ and $c_1$ subsumes $s_1$ in $\mathcal{O}_1$, then we discard the

mapping with the lower prediction score. The remaining mappings are denoted as $\mathcal{M}'_p$.

$\mathcal{M}'_p$ is merged with the seed mappings $\mathcal{M}_s$ to give the final ensemble output: $\mathcal{M} = \mathcal{M}'_p \cup \mathcal{M}_s$. Note that although the seed mappings are used as positive samples for training, it is still possible that some of them will have low prediction scores as embedding, learning and prediction is a probabilistic procedure.

## 5   Evaluation

### 5.1   HeLiS and FoodOn

**Experiment Setting** We first evaluate the ML extension[10] for aligning HeLiS and FoodOn, where we augment LogMap[11] by using its anchor mappings for the seeds. The augmented system is denoted as LogMap[anc]-ML. It is compared with the original LogMap and another SOTA system AML v3.1[12] which has been highly ranked in many OAEI tasks [10].

In order to precisely assess an alignment of two ontologies we would need a set of gold standard (GS) mappings against which to measure precision and recall. This is typically not available due to the cost of checking each of a potentially very large number of possible mappings. In the case of HeLiS and FoodOn we have a partial GS consisting of 372 mappings obtained by manually checking a much larger set of candidate mappings computed by LogMap; however, this is still (highly) incomplete, and clearly biased towards mappings that can be found using the techniques employed in LogMap. Therefore, besides the recall of this partial GS (denoted as Recall[GS]), we have computed approximate precision and recall (denoted as Precision[≈] and Recall[≈], respectively) as follows.

First, given a (possibly empty) set of GS mappings $G$ and a set of computed mappings $M$, we estimate Precision[≈] for $G$ and $M$ (denoted Precision$^{\approx}_{G,M}$) by selecting at random a set $S \subseteq M \setminus G$ and manually checking the mappings in $S$ to identify the set $S_v \subseteq S$ of valid mappings. We then compute:

$$\text{Precision}^{\approx}_{G,M} = \frac{TP_{G,M}}{|M|} = \frac{|M \cap G| + \frac{|S_v|}{|S|} \times |M \setminus G|}{|M|}, \qquad (2)$$

where $|\cdot|$ denotes set cardinality, and $TP_{G,M}$ represents the approximate number of *true positive* mappings. Note that, if $G = \emptyset$ (*i.e.*, no gold standard is available), then this becomes simply $|S_v|/|S|$; if $M \setminus G = \emptyset$ (*i.e.*, all the output mappings are among the GS), then this becomes precision w.r.t. the GS. For the recall, we estimate the total number of valid mappings using the GS as well as the union of the output mappings of all available systems (*i.e.*, LogMap[anc]-ML, LogMap and AML), which we denote $M'$; then, for a given system that computes a set

---

[10] Codes: `https://github.com/KRR-Oxford/OntoAlign/tree/main/LogMap-ML`

[11] `https://github.com/ernestojimenezruiz/loap-matcher`

[12] `https://github.com/AgreementMakerLight/AML-Project`

| Method | Mappings # | Precision$^\approx$ | Recall$^\approx$ | F1$^\approx$ | Recall$^{\text{GS}}$ |
|---|---|---|---|---|---|
| LogMap (anchor mappings) | 311 | **0.887** | 0.278 | 0.423 | 0.602 |
| LogMap | 417 | 0.676 | 0.284 | 0.400 | 0.712 |
| AML | 544 | 0.636 | 0.349 | 0.451 | 0.694 |
| LogMap$^{\text{anc}}$-ML (no ensemble) | 1154 | 0.675 | 0.785 | 0.726 | 0.806 |
| LogMap$^{\text{anc}}$-ML | 1207 | 0.685 | **0.833** | **0.752** | **0.839** |

Table 1: The results of aligning HeLiS and FoodOn

of mappings $M$, we estimate the recall of the system to be:

$$\text{Recall}^\approx = \frac{TP_{G,M}}{|G| + \frac{|S'_v|}{|S'|} \times |M' \setminus G|}, \tag{3}$$

where $S'$ denotes a random set from $M' \setminus G$ and $S'_v \subseteq S'$ are the valid mappings in $S'$ (by manual checking). We further calculate an approximate F1 Score:

$$\text{F1}^\approx = \frac{2 \times \text{Precision}^\approx \times \text{Recall}^\approx}{\text{Precision}^\approx + \text{Recall}^\approx}. \tag{4}$$

The settings of LogMap$^{\text{anc}}$-ML are adjusted by optimizing the result on the validation mapping set $\mathcal{M}_s^v$, which consists of 10% of all the seed mappings, and the results in Table 1 are based on these optimized settings. For the validation results of different settings, please see the ablation study in Section 5.3.

**Results** In Table 1 we can see that LogMap$^{\text{anc}}$-ML without the ensemble to the seed mappings outputs 1154 mappings — more than twice as many as the original LogMap and AML — while the ensemble (*i.e.*, including the seed mappings) adds 53 more mappings. LogMap$^{\text{anc}}$-ML has much higher recall than AML and LogMap; for example, Recall$^\approx$ is increased from 0.284 to 0.833 when the ML extension is added to LogMap. This is consistent with our assumption that SiamNN together with the OWL2Vec$^*$ embedding can consider more additional contextual information and word semantics of two to-be-mapped classes. Meanwhile, LogMap$^{\text{anc}}$-ML (no ensemble) has similar precision to LogMap and 6.1% higher precision than AML, while the ensemble with the seed mappings further improves precision from 0.675 to 0.685. As a result, F1$^\approx$ of LogMap$^{\text{anc}}$-ML is 88.3% higher than LogMap and 67.0% higher than AML.

## 5.2   OAEI Conference Track

**Experiment Setting** We also evaluated our ML extension on all the 21 class alignments of the 17 ontologies of the OAEI conference track,[13] where the open reference *ra1* with 259 class mappings (*i.e.*, the GS) is adopted to calculate the standard precision, recall and F1 score. Note that we merge the output mappings

---

[13] http://oaei.ontologymatching.org/2020/conference/

| Method | Mappings # | Precision | Recall | F1 Score |
|---|---|---|---|---|
| StringEquiv | 148 | **0.935** | 0.498 | 0.650 |
| AML | 223 | 0.803 | 0.691 | 0.743 |
| SANOM[oaei] | 252 | 0.778 | 0.757 | 0.767 |
| Wiktionary[oaei] | 184 | 0.821 | 0.583 | 0.682 |
| VeeAlign[oaei] | 253 | 0.791 | 0.772 | **0.781** |
| DeepAlignment | – | 0.710 | **0.800** | 0.750 |
| LogMap[anc] | 139 | 0.892 | 0.479 | 0.629 |
| LogMap[anc]-ML | 157 | 0.917 | 0.555 | 0.691 |
| LogMap | 190 | 0.842 | 0.618 | 0.713 |
| LogMap-ML | 190 | 0.881 | 0.645 | 0.745 |
| LogMap[oaei] | 198 | 0.843 | 0.645 | 0.731 |
| LogMap[oaei]-ML | 197 | 0.875 | 0.665 | 0.756 |
| AML[oaei] | 220 | 0.827 | 0.703 | 0.760 |
| AML[oaei]-ML | 222 | 0.842 | 0.723 | 0.778 |

Table 2: The results of the OAEI conference track.

of the 21 alignments of each system and compare all the mappings to the GS to directly calculate these metrics. The results are shown in Table 2. As the output mappings of AML and LogMap available on the OAEI website of 2020 are slightly different from those generated by our local running of LogMap and AML v3.1 (perhaps this is due to different parameter settings or versions), we report both results, where the former is denoted by the superscript 'oaei'.

Instead of processing each ontology alignment independently, we merged the seed mappings of all the 21 alignments to train one prediction model and applied this model to predict the candidate mappings of all the alignments. We did not use the LogMap over-estimation mappings but adopted all the cross-ontology class pairs of each alignment, as the total number $(98, 688)$ did not cause scalability issues. As well as LogMap[anc]-ML, we also report the results of LogMap-ML and LogMap[oaei]-ML, which adopt the corresponding LogMap output mappings for the seeds. To show the generality, we also applied the ML extension to AML, which we denote AML[oaei]-ML. Note that the output mappings of LogMap and AML are adopted for training because they have high precision in this case. The reported results of all the ML extensions are based on the class input embedded by the pre-trained OWL2Vec* and the SiamNN with MLP. The baselines include four classic systems (*i.e.*, Wiktionary [28], SANOM [24], LogMap and AML), two SOTA ML-based systems (*i.e.*, VeeAlign [15] and DeepAlignment [19]), and StringEquiv which labels a mapping as true if its class names are the same (case insensitive). Note precision and recall are based on the average of several repetitions of training and prediction, while F1 score is calculated with the averaged precision and recall. The DeepAlignment result is from its paper [19].

**Results** From Table 2 we can first confirm the observation from HeLiS and FoodOn, *i.e.*, that the ML extension can improve both the recall and precision

of the original alignment system: the F1 score is improved by 9.9%, 4.3%, 3.4% and 2.4% for LogMap[anc], LogMap, LogMap[oaei] and AML[oaei] respectively. The improvements are not as large as in aligning HeLiS and FoodOn because the to-be-aligned conference ontologies are much smaller and less complex, and the original systems have already been highly optimized for these alignments. Second, our augmented method LogMap[oaei]-ML is a bit worse than VeeAlign[oaei][14] and quite competitive to SANOM[oaei], w.r.t. the F1 score. Note these two systems are ranked in the first positions in the 2020 and 2019 rankings, respectively. LogMap[oaei]-ML also has a slightly higher F1 score than DeepAlignment. Meanwhile, the F1 score of the augmented AML, *i.e.*, AML[oaei]-ML is very competitive to VeeAlign[oaei] and is better than the other baselines (*e.g.*, 1.4% higher than SANOM[oaei]). As the ML extension is trained based on the confident seed mappings from the original systems, the augmented AML and LogMap have much higher precision than VeeAlign[oaei], SANOM[oaei] and DeepAlignment.

### 5.3   Ablation Study

**Experiment Setting** We present the ablation study of the prediction model with different embedding and neural network settings. To this end, standard precision, recall, F1 score and accuracy of the trained model on the validation mapping set are reported. The threshold $\theta$ is searched from 0 to 1 with a step of 0.02, and the reported results are based on the threshold that leads to the best F1 score. All the four metrics are calculated by averaging the results of several repetitions of training and validation. Note the model with the best F1 score and its associated optimum $\theta$ are adopted in calculating the final output mappings as evaluated in Section 5.1. As the validation set, especially its generated negative mappings, are quite simple in comparison with the candidate mappings for prediction, the validation results in Table 3 are much better than the final results in Table 1, but this does not impact our validation of different settings.

We evaluated *(i)* Word2Vec which was trained with a corpus of Wikipedia articles from 2018, OWL2Vec* without pre-training and OWL2Vec* pre-trained with the above Wikipedia corpus; *(ii)* different networks including the SiamNNs, and the original networks (MLP, BiRNN and AttBiRNN) for which the two input vectors are concatenated; and *(iii)* the class vs the path as the input. The dimensions of Word2Vec, OWL2Vec* and the pre-trained OWL2Vec* are set to 200, 100 and 200 respectively. The hidden neural sizes of MLP and BiRNN are both set to 200, while the attention size of AttBiRNN is set to 50. The epoch number and the batch size are set to 14 and 8 in training.

**Results** On the one hand we find OWL2Vec* with pre-training has better performance than the original Word2Vec and OWL2Vec* without pre-training; for example, the best F1 scores of these three settings are 0.927, 0.903 and 0.911 respectively. This observation, which is consistent under different settings, is as expected because the pre-trained OWL2Vec* incorporates words' common sense

---

[14] VeeAlign has been tailored to the Conference and Multifarm OAEI tracks [15].

| Embedding (Class) | Neural Network | Precision | Recall | F1 Score | Accuracy |
|---|---|---|---|---|---|
| | MLP | 0.809 | 0.798 | 0.803 | 0.869 |
| | BiRNN | 0.741 | 0.940 | 0.827 | 0.869 |
| Word2Vec | AttBiRNN | 0.790 | 0.941 | 0.859 | 0.897 |
| | SiamNN (MLP) | 0.874 | 0.941 | 0.903 | 0.932 |
| | SiamNN (BiRNN) | 0.808 | **0.952** | 0.874 | 0.909 |
| | SiamNN (AttBiRNN) | 0.828 | **0.952** | 0.884 | 0.917 |
| | MLP | 0.769 | 0.869 | 0.815 | 0.869 |
| OWL2Vec* | BiRNN | 0.751 | 0.929 | 0.830 | 0.873 |
| (Without | AttBiRNN | 0.708 | 0.976 | 0.820 | 0.857 |
| Pre-training) | SiamNN (MLP) | 0.854 | 0.976 | 0.911 | 0.936 |
| | SiamNN (BiRNN) | 0.924 | 0.833 | 0.874 | 0.921 |
| | SiamNN (AttBiRNN) | 0.829 | 0.905 | 0.862 | 0.901 |
| | MLP | 0.826 | 0.845 | 0.835 | 0.889 |
| OWL2Vec* | BiRNN | 0.821 | 0.905 | 0.859 | 0.901 |
| (With | AttBiRNN | 0.828 | 0.860 | 0.842 | 0.893 |
| Pre-training) | SiamNN (MLP) | **0.952** | 0.905 | **0.927** | **0.952** |
| | SiamNN (BiRNN) | 0.914 | 0.881 | 0.897 | 0.933 |
| | SiamNN (AttBiRNN) | 0.854 | 0.893 | 0.871 | 0.913 |

Table 3: The results over the validation mapping set of different embedding and network settings for aligning HeLiS and FoodOn. Class embedding is adopted as the input.

semantics and local context in the ontology. On the other hand we can find the SiamNNs outperform their original networks, and the SiamNN with MLP achieves the best performance. The former validates that the SiamNN architecture, which can align two embedding spaces according to the given training mappings, is more suitable for this task. The latter means further feature learning by RNN over the embedding makes no additional contribution in comparison with MLP. We also validated the above networks using path embedding as input, and the results are worse than their correspondences using the class embedding as input; for example, the best validation F1 score is 0.885 which is worse than 0.927 in Table 3. This may be due to the fact that the relevant predictive information from the class subsumers has already been encoded by OWL2Vec*.

## 6   Conclusion and Discussion

In this paper we presented a general ML extension to existing ontology alignment systems such as LogMap and AML. Briefly, it first adopts the confident mappings from an original system, such as the anchor mappings of LogMap, as well as some external class disjointness constraints, to generate training samples, then uses an ontology tailored language model OWL2Vec* and a SiamNN to train a model and predict the candidate mappings, and finally filters out invalid mappings according to their predicted scores and a subsumption-based logical assessment. According to the evaluation on an industrial use case and the alignments of the

OAEI conference track, the ML extension is shown to be effective in improving both precision and recall. We discuss below some more subjective observations and possible directions for future work.

**Running Time** Computation of the ML extension mainly lies in training and validation. With a laptop equipped by 2.3 GHz Intel Core i5 and 16 GB memory, the training of the 6 networks as set out in Section 5.3 with the class input using the pre-trained OWL2Vec$^*$ and the seeds from LogMap anchors takes 1.3 minutes for HeLiS and FoodOn, and 1.1 minutes for the OAEI conference track. It is possible to achieve even better matching performance via an exhaustive exploration of more settings, *e.g.*, the network hidden layer size, but this requires significantly more computation.

**Seed Mappings** In distant supervision we assume that the seed mappings used for training are precise. In aligning HeLiS and FoodOn, we also considered extracting the seed mappings from the output mappings of LogMap and AML, both of which have a larger size but lower precision (0.676 and 0.636 respectively) compared to the LogMap anchors. However both lead to lower precision and recall, and the approximate F1 score drops to 0.657 and 0.696 respectively. Class disjointness constraints are also important for filtering out false-positive mappings and generating high precision seeds mappings; in the future work we plan to study semi-automatic neural-symbolic methods for deriving robust class disjointness constraints.

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
