# OpenReview forum: "Augmenting Ontology Alignment by Semantic Embedding and Distant Supervision"
_eswc-conferences.org/ESWC/2021/Conference/Research_Track — ESWC 2021 Research_

### Official Review · AnonReviewer5 · 2021-01-11
**A new ML extension based on embeddings and Siamese NNs to enhance ontology alignment systems.**

**Rating:** 2
**Confidence:** 4
**Impact:** 4
**Design And Technical Quality:** 4

**Review:**

This paper presents a new embedding and neural network based method to enhance conventional ontology alignment systems such as AML and LogMap. The method learns ontology embeddings by OWL2Vec and uses a SiamNN to capture ontology alignment. Experiments on OAEI benchmarks demonstrate the effectiveness of the method.

Pros:

1) It is solid work. The proposed method combines the advantages of ML techniques (e.g., semantic representations, alignment learning and prediction) and traditional systems (blocking,  high-confidence seed mapping initialization) to get ensemble alignment. It seems to be a meaningful attempt to the ontology alignment problem.

2) Authors conduct sufficient ablation studies to demonstrate the effectiveness of the OWL embeddings and pre-training strategy, providing many useful findings for future work.

3) The paper is well structured , clearly written, and easy to follow.

Cons:

1) It seems that the datasets used in the experiments are somewhat small. It would be interesting to see the results on much larger ontology alignment datasets, such as Gene ontologies.

Overall, I appreciate this hybrid ontology alignment system based on both embedding and traditional techniques. I recommend accepting this submission.

I acknowledge that I have read the above response.

**Anonymity:**

Yes, I would like my review to remain anonymous.

**Reuse And Availability:**

4: High

**Strong Points:**

1) This paper presents a solid method that combines the advantages of ML and traditional techniques for ontology alignment. It is a meaningful attempt to the ontology alignment problem.

2) Experiments demonstrate the effectiveness of this hybrid method. The paper shows that embedding-based and traditional methods can enhance each other and output ensemble ontology alignment.


**Subreviewer:**

I delegated this review to a subreviewer.

**Weak Points:**

1)  It would be interesting to see the results of large-scale ontology alignment.

---

> ### Author Rebuttal · Authors · 2021-01-28
>
> We would like to thank the reviewer for the provided comments.
>
> Regarding evaluation on more large scale matching tasks, we agree that this will be beneficial and we will continue to extend the range of evaluation tasks in our future work. For this purpose we will consider the OAEI Large BioMed track and the OAEI Disease and Phenotype track as well as other industrial ontology alignment tasks from industry partners. We also plan to participate in the next OAEI campaign.

---

### Official Review · AnonReviewer2 · 2021-01-14
**Important contribution to machine learning-based ontology alignment with a novel, innovative method**

**Rating:** 3
**Confidence:** 4
**Impact:** 4
**Design And Technical Quality:** 5

**Review:**

This paper proposes a machine learning-based extension of traditional ontology alignment approaches, such as LogMap and AgreementMakerLight. In the paper, the extension is evaluated on LogMap combined with distant supervision, ontology embeddings (OWL2Vec) and a Siamese Neural Network, which equips the alignment approach with considerably richer semantics. Its evaluation is performed on the alignment of two food ontologies as well as a standard OAEI task and it is reported to strongly boost recall while at the same time achieving small improvements in precision.

Building on a very profound presentation of the background of the field and its variations of machine learning-based approaches, this paper presents an innovative and novel method to improve ontology alignment. Given the extensive evaluation and substantial improvement in recall for real-world settings, it provides a valuable contribution to the field. Especially a very extensive ablation study and an impeccable argumentation solidifies the validation of choices made in the approach.

While I appreciate the fact that such a tremendously higher recall is invaluable for real world applications of ontology learning, I was wondering whether you have any intuition on why the precision of LogMap with anchor mappings is so much higher than with the ML extension in the food ontology use case. It seems not to be recurring behavior in the OAEI case. Additionally, it would be great if  the best results in each table could be highlighted. Another question that came to mind is the difference between averaging and concatenation of vectors. It seems that concatenation worked better, since this setting was reported, but it would be interesting to know the extend of the difference.

**Anonymity:**

Yes, I would like my review to remain anonymous.

**Reuse And Availability:**

4: High

**Strong Points:**

The paper presents an interesting and novel idea in combining several ML-based extensions to LogMap. Besides a strong motivation for this kind of approach, the proposal is very well embedded within its relevant background. Argumentation, style and structure of the paper are at a very high level, equipped with illustrative, excellent examples. The paper is well within the scope of the chosen topic and promises to provide for interesting discussions at the conference.


**Subreviewer:**

I submitted this review.

---

> ### Author Rebuttal · Authors · 2021-01-28
>
> We would like to thank the reviewer for the provided comments.
>
> 1. The LogMap anchor mappings are those mappings with a high lexical matching degree, and they have a high precision but a very low recall (cf. Section 2.1). LogMap with ML extension learns a model from seed mappings (extracted from anchor mappings) and then predicts other mappings. The learning and prediction is uncertain, leading to additional errors, and it also propagates errors (false seed mappings).  Thus LogMap with ML extension has lower precision. Note learning and prediction leads to much higher recall, and as a result has higher F1 score. We will clarify this in the final paragraph of Section 5.1.
>
> 2. It is a very good suggestion to highlight the best results in each column or row. We will include in the revised version.
>
> 3. In embedding a path, we concatenated the vectors of its classes. With a suitable ML model, the concatenation operator would perform better as it keeps all the information as the input. We will consider the ablation study of comparing the concatenation operator and the averaging operator. Regarding our prediction model, we adopt the SiamNN which has two inputs -- the vectors of the two to-be-matched classes. While for the baseline prediction models such as MLP which has one input, we adopt the concatenation of the two vectors.

---

> > ### Comment · AnonReviewer2 · 2021-01-29
> > **Response to rebuttal Paper 41**
> >
> > Thank you for your detailed explanations and comments.

---

### Official Review · AnonReviewer1 · 2021-01-15
**A technical contribution to combining embedding with traditional ontology matching approaches, whereas the evaluation is limited.**

**Rating:** 1
**Confidence:** 5
**Impact:** 3
**Design And Technical Quality:** 4

**Review:**

The paper presents to add embedding-based training and prediction after existing ontology matching (OM) systems so as to recall more mappings and improve the precision. Based on the mappings from state-of-the-art OM systems LopMap or AML, an ontology embedding model is used together with a neural network to train low-dimensional vectors for classes and the similarity is computed based on the distance of vectors in numerical space. The evaluation of the approach is conducted on matching two large food ontologies, where the recall has largely increased and the precision also improved a little in comparison with LogMap and AML. A further evaluation on the OAEI (yearly ontology matching competition since 2004) Conference Track shows the improvements as well. As the embedding-based techniques have made great progress in many disciplines including natural language processing and knowledge graph, the OM community started to see the power of embedding in recent years. It's meaningful and interesting to explore what the combination of traditional approaches with embedding would bring about. The model devised in the paper is elaborated with reasonable considerations and technical details. The paper is clearly written and easy to follow. I think it's a solid technical contribution that can benefit the ESWC audience.

The main problem of the paper lies in the limited evaluation. To demonstrate a general-purpose extension added to any OM system,  the success on solely one large matching task does not seem to be enough. The OAEI conference tasks are small-scaled  and the improvement the presented model made is limited. The embedding-based techniques tend to favor recall so the recall improvement is understandable. On the other hand, it's harder for embeddings to produce precise results, and the performance of neural networks often depends on layer and parameter settings and choices from many options. All these require extensive empirical studies so as to convince the applicability of the presented model to any existing OM system.

Some other problems:
- The mappings from an existing OM system are used as seeds in the paper and their correctness seems vital for the final quality of matching. How to ensure the precision of the seed mappings should be addressed; and more importantly, what portion/size of the seeds is necessary to initiate the model to work?
- When matching two food ontologies where one has a lot of instances and much fewer classes and the other many more classes and fewer instances, instances in the former are transformed into classes and accordingly the instanceOf relation is changed to subClassOf. The paper simply says "in order to facilitate alignment". This definitely needs some clarification, as such an transformation totally changes the semantics of the ontology, and thus the resultant mappings, which then affect their applications and so on.




**Anonymity:**

Yes, I would like my review to remain anonymous.

**Reuse And Availability:**

4: High

**Subreviewer:**

I submitted this review.

---

> ### Author Rebuttal · Authors · 2021-01-28
>
> We would like to thank the reviewer for the provided comments.
>
> 1. Regarding evaluation on more large scale matching tasks, we agree that this will be beneficial and we will continue to extend the range of evaluation tasks in the near  future. Note that to show its generality, we have applied our extension to more than one OM systems (cf. Table 1 and Table 2). We also plan to participate in the next OAEI evaluation campaign.
>
> 2. Regarding ensuring the precision of the seed mappings, we used the most confident mappings from the traditional systems and further filtered them using cross-ontology class disjointness constraints. See the first paragraph in Section 4.1 for more details. The higher precision, the better. A larger number of seed mappings would lead to a more robust model. However, the representativity/diversity of the seed mappings is more important. As part of our future work we plan to investigate the use of consensus mappings (i.e., mappings proposed/voted by different systems) as well as heuristic rules and human interaction strategies to obtain more diverse seed mappings.
>
> 3. We agree that the transformation from instances to classes does change the semantics, although in some cases there is a thin line among classes and instances and it depends on the modelling choices. Furthermore,  our industry partner believes that ontology integration with this transformation can still support its application in constructing a food knowledge graph. The resulting ontologies are in any case useful for comparing the different alignment systems. We will clarify this and provide more details on the instance to class transformation.

---

> > ### Comment · AnonReviewer1 · 2021-01-30
> > **The review comments agreed by the authors.**
> >
> > The authors basically agreed on my review comments. Regarding Point 3 in their rebuttal, please go ahead clarify in the paper. Regarding point 2, please add in the paper about the seeds in terms of the-more-the-better as well as the diversity issue. Regarding Point 1, I insist that evaluating on solely one LARGE-SCALED matching task (Table 1) is not satisfactory, as conference tasks are at most medium-scaled (Table 2). It's good to know that the authors plan to extend empirical studies and participate in OAEI.

---

### Official Review · AnonReviewer3 · 2021-01-18
**The paper proposes an interesting solution providing some promising preliminary results but, from time to time it lacks of clarity. Detailed comments are reported in the following.**

**Rating:** 1
**Confidence:** 3
**Impact:** 3
**Design And Technical Quality:** 3

**Review:**

The paper proposes a solution grounded on extending existing ontology alignment with a module for predicting candidate/valid ontology mappings. Starting from existing mappings computed by systems at the state of the art and some disjointness constraints for generating training samples, an ontology tailored language model OWL2Vec and a SiamNN to train a model and predict the candidate mappings are used. Finally, invalid mappings are filtered out according to predicted scores and a subsumption-based logical assessment. An experimental evaluation on the HeLiS and FoodOn ontologies as well as on ontologies of the OAEI conference track is provided.

The paper proposes an interesting solution providing some promising preliminary results but, from time to time it lacks of clarity. Detailed comments are reported in the following.

A more clear and detailed discussion concerning the limitations of the methods at the state of the art and the advances/value added of the proposed solution should be provided, whilst section 2.1 could be made definitely more short or even deleted.

A large part of Section 3 is not really useful to the purpose of the paper. However, by the end of this section it is reported "Classic systems often fail to identify such errors, even when using logical assessment (as in LogMap) due to missing class disjointness axioms in the source ontologies." This part should be extended and clarified, since it represents an improvement over the state of the art that the proposed solution. Specifically, it seems that here, a general issue that the proposed solution is able to manage (differently from the system at the state of the art) seems to be introduced. However, such general issue should be clearly presented whilst currently there is only a quick mention by the use of a short example.

Section 4 starts with the sentence "We will present our ML extension w.r.t. LogMap, but it can be used with any "traditional" system that is capable of generating high precision mappings for use in the training phase (in our evaluation we use AML as well as LogMap)."However, a clear introduction on when the training phase occurs and what is the purpose is missing. Actually, a clear presentation of the overall process and goal should be given, hence details could be provided. At this regards, the text "As shown in Fig. 3, the ML extension comprises ... to reduce the search space." is appropriate and appreciated but the goal is still missing.

Disjointness constraints do not seem a secondary aspect of the proposed solution. On page 7, a rough proposal on how to assess such constraints in a way that does not necessarily require human intervention is sketched. However, its impact on the overall results should be somehow evaluated, given that the experiments mostly rely on disjointness constraints that have been manually defined.

Furthermore, it seems that seed mappings are actually the mappings remaining in M_a after applying/verifing disjointness constraints. If so, this should be presented in a more clear way. I would suggest, to start with a clear presentation of what the seed mapping are and how they are obtained, hence to explain how disjointness constraints are build.

More details should be provided for OWL2Vec* and its role.

MINOR:

- Page 1: "Ontology alignment has been been investigated for many years." --> "Ontology alignment has been investigated for many years."

AFTER REBUTTAL
-  I acknowledge that I have read the response from the authors

**Anonymity:**

Yes, I would like my review to remain anonymous.

**Reuse And Availability:**

3: Medium

**Strong Points:**

Interesting solution
promising experiemtnal results

**Subreviewer:**

I submitted this review.

**Weak Points:**

- lack of clarity
- the current status of the paper remids more a workshop paper.

---

> ### Author Rebuttal · Authors · 2021-01-28
>
> We would like to thank the reviewer for the provided comments.
>
> 1. Regarding Section 2.1 on the background of LogMap, we intended to make the paper self-contained as LogMap is selected as the main reference system for the experiments, but we will try to compress it if we need additional space to address the provided suggestions.
>
> 2. Regarding Section 3, we aim at introducing the industrial case study of aligning two food ontologies. This is different from the widely used OAEI tasks. We agree that the data introduction can be moved to the evaluation section, but we believe that the introduction to the new challenges in this case study is beneficial to motivate our method. For the final statement on class disjointness, this refers to the disjointness between classes across ontologies and few current systems have utilized such disjointness. We have introduced why this challenge exists in this case study as the two to-be-aligned ontologies model the domain in a different way and lexically similar concepts are classified in branches that are implicitly disjoint. We will clarify the final sentence in the revised version.
>
> 3. Regarding the first paragraph in Section 4, we will reorganize it by first introducing the general workflow, and then mentioning the generality w.r.t. other traditional systems. This should address the issue as to “when the training phase occurs”. We will also clarify the objective of the method.
>
> 4. Regarding the disjointness constraints, we plan to study, implement and evaluate the sketched solution for discovering cross-ontology class disjointness constraints (possibly with human interaction) in the future work.  Our current work shows that even a few manually defined disjointness constraints have a significant positive impact.
>
> 5. Regarding the seed mapping, we will reorganize its presentation by first presenting its definition and then its calculation.
>
> 6. Regarding the role of OWL2Vec*, OWL2Vec* is used to compute the vector embeddings (features) of ontology classes. The machine learning models receive as input the vectors of the classes involved in an alignment.

---

### Decision · Program_Chairs · 2021-02-23

**Decision:**

Accept

**Comment:**

The paper proposes a machine learning-based extension of traditional ontology alignment approaches. This is an interesting work addressing an important problem. All the reviewers are positive and are satisfied with the authors‘ response to their criticisms, thus we recommend acceptance.

To address the comments of the reviewers the authors are requested to expand the discussion concerning limitations of the state-of-the-art methods in the final version of the paper.